Molecular typing and profiling of topoisomerase mutations causing resistance to ciprofloxacin and levofloxacin in Elizabethkingia species

Jian Ming-Jr 1 2
Cheng Yun-Hsiang 1 2
Perng Cherng-Lih 1 2
Shang Hung-Sheng 1 2 iamkeith@mail.ndmctsgh.edu.tw
1 Graduate Institute of Medical Science, National Defense Medical Center , Taipei , Taiwan
2 Division of Clinical Pathology, Department of Pathology, Tri-Service General Hospital, National Defense Medical Center , Taipei , Taiwan
Tulkens Paul
Electronic publication date: 2018 Sep 12
Publication date: 2018
Volume: 6
Electronic Location ID: e5608
Received 2018 Jun 5; Accepted 2018 Aug 19
Copyright: © 2018 Jian et al.
Copyright year: 2018
Copyright holder: Jian et al.
License: This is an open access article distributed under the terms of the Creative Commons Attribution License, which permits unrestricted use, distribution, reproduction and adaptation in any medium and for any purpose provided that it is properly attributed. For attribution, the original author(s), title, publication source (PeerJ) and either DOI or URL of the article must be cited.
License URL: https://creativecommons.org/licenses/by/4.0/

Keywords: Elizabethkingia spp., Fluoroquinolone resistance, gyrA, Molecular typing, High-resolution melting

Funding: Tri-Service General Hospital, Taipei, Taiwan, ROC, Grant Numbers TSGH-C104-203 and TSGH-C106-170 This study was supported by Tri-Service General Hospital, Taipei, Taiwan, ROC, Grant Numbers: TSGH-C104-203 and TSGH-C106-170. The funders had no role in study design, data collection and analysis, decision to publish, or preparation of the manuscript.

==============================
Objectives

Several Elizabethkingia species often exhibit extensive antibiotic resistance, causing infections associated with severe morbidity and high mortality rates worldwide. In this study, we determined fluoroquinolone susceptibility profiles of clinical Elizabethkingia spp. isolates and investigated the resistance mechanisms.

Methods

In 2017–2018, 131 Elizabethkingia spp. isolates were recovered from specimens collected at tertiary care centers in northern Taiwan. Initial species identification using the Vitek MS system and subsequent verification by 16S rRNA sequencing confirmed the presence of Elizabethkingia anophelis (n = 111), E. miricola (n = 11), and E. meningoseptica (n = 9). Fluoroquinolone susceptibility was determined using the microbroth dilution method, and fluoroquinolone resistance genes were analyzed by sequencing.

Results

Among Elizabethkingia spp. isolates, 91% and 77% were resistant to ciprofloxacin and levofloxacin, respectively. The most prevalent alterations were two single mutations in GyrA, Ser83Ile, and Ser83Arg, detected in 76% of the isolates exhibiting fluoroquinolone MIC between 8 and 128 μg/ml. Another GyrA single mutation, Asp87Asn, was identified in two quinolone-resistant E. miricola strains. None of the isolates had alterations in GyrB, ParC, or ParE. We developed a high-resolution melting assay for rapid identification of the prevalent gyrA gene mutations. The genetic relationship between the isolates was evaluated by random amplified polymorphic DNA PCR that yielded diverse pulsotypes, indicating the absence of any temporal or spatial overlap among the patients during hospitalization.

Conclusion

Our analysis of fluoroquinolone-resistant Elizabethkingia spp. isolates provides information for further research on the variations of the resistance mechanism and potential clinical guidance for infection management.

Introduction

The genus Elizabethkingia has been recently revised to include several species based on whole-genome sequencing analysis (Doijad et al., 2016; Nicholson et al., 2017). Elizabethkingia species are non-motile, non-fastidious, and glucose-non-fermentative gram-negative bacilli (Janda & Lopez, 2017). Three species, Elizabethkingia meningoseptica, E. miricola, and E. anophelis, are known to cause diseases in humans (Green, Murray & Gea-Banacloche, 2008; Jean et al., 2014; Lau et al., 2016). Recent studies suggest that certain strains causing sporadic cases of meningitis and bacteremia, previously identified as E. meningoseptica, belong to E. anophelis (Chew et al., 2017; Lin et al., 2017). Several outbreaks of E. anophelis-associated infections have been reported, including two outbreaks in the US Midwest in 2016 with 65 confirmed cases (Coyle, 2017; Janda & Lopez, 2017). Elizabethkingia species also cause outbreaks in intensive care units as emerging pathogens of nosocomial infections with a high mortality rate and severe morbidity in critically ill patients (Jean et al., 2014; Lau et al., 2016; Opota et al., 2017; Hu et al., 2017). Because of their ability to accumulate different resistance mechanisms and a growing number of more vulnerable hosts, the prevalence of multidrug-resistant Elizabethkingia species has increased in the past decades, limiting the options for treatment (Janda & Lopez, 2017; Jean et al., 2014). For instance, resistance to carbapenems is mediated by metallo-β-lactamases (Breurec et al., 2016; Chen et al., 2017; Colapietro et al., 2016). A previous report indicated that fluoroquinolones are suitable for treating E. meningoseptica bacteremia (Huang, Lin & Wang, 2018), and empirical evidence indicates that they are effective in treating E. anophelis and E. miricola infections (Coyle, 2017; Figueroa Castro et al., 2017; Green, Murray & Gea-Banacloche, 2008; Zdziarski et al., 2017). However, a detailed analysis of fluoroquinolone-resistant Elizabethkingia spp. infections has not yet been performed.

Fluoroquinolones, such as ciprofloxacin (CIP) or levofloxacin (LVX), have two bacterial drug targets, DNA gyrase and DNA topoisomerase IV (Khodursky, Zechiedrich & Cozzarelli, 1995; Kreuzer & Cozzarelli, 1979). Each enzyme is a heterotetramer, with gyrase composed of two GyrA and two GyrB subunits and topoisomerase IV composed of two ParC and two ParE subunits. Mechanisms of fluoroquinolone resistance include mutational alterations in drug target affinity, increased efflux pump expression, and acquisition of resistance-conferring genes (Hooper & Jacoby, 2016). Single amino acid changes in either gyrase or topoisomerase IV can cause quinolone resistance. In gram-negative bacilli, mutations have been typically localized to the amino-terminal region of the primary target, GyrA (Yoshida et al., 1990), a region conserved among all potential quinolone targets. Mutations in these conserved regions of GyrB, ParC, and ParE are also known to confer fluoroquinolone resistance, like the amino-terminal GyrA region (Heisig, 1996; Yoshida et al., 1991). Accordingly, the genomic DNA regions encoding the conserved protein regions of GyrA, GyrB, ParC, and ParE have been termed quinolone resistance-determining regions (QRDRs).

In this study, we aimed to assess the relationship between the quinolone-resistant phenotype of clinical Elizabethkingia spp. isolates in Taiwan and mutations in their DNA gyrase and DNA topoisomerase IV genes.

Materials and Methods

Bacterial isolates

In 2017–2018, 131 isolates of Elizabethkingia spp. (E. anophelis, n = 111; E. meningoseptica, n = 9; E. miricola, n = 11) were recovered by bacterial culture from respiratory tract, urine, catheter tip, and blood specimens collected at the Tri-Service General Hospital, tertiary care centers in northern Taiwan. The species were initially identified using the Vitek MS system with the IVD 3.0 database (BioMérieux, Marcy-l’Étoile, France). Isolates identified as Elizabethkingia species using a previously reported study (Cheng et al., 2018). Briefly, MALDI-TOF spectral analysis software identified significant species-specific peaks to create reference masses for efficient and accurate identification of Elizabethkingia spp. All bacterial isolates were kept frozen until used in this study.

Antimicrobial susceptibility

MIC of CIP and LVX were determined using the broth microdilution method. The susceptibilities were evaluated according to guidelines published by the Clinical and Laboratory Standards Institute (CLSI) including antibiotic-specific breakpoints (CIP: susceptible ≤1 μg/ml, resistant ≥4 μg/ml; LVX: susceptible ≤2 μg/ml, resistant ≥8 μg/ml).

DNA extraction

Genomic DNA was isolated using a previously reported protocol (Syn & Swarup, 2000). Briefly, cellular lysis is achieved by a combination of EDTA/SDS detergent lysis and brief heat treatment. An additional phenol/chloroform step further deproteinates the preparation yielding DNA of good quality. Using a picodrop spectrophotometer, purified genomic DNA concentrations were determined by measuring the optical density at 260 nm, whereas the purity was estimated by calculating the ratio of the optical densities measured at 260 and 280 nm. DNA samples were stored at −20 °C until PCR was performed.

Bacteria species identification by 16S rRNA sequencing

The microbial identification accuracy was verified by 16S rRNA sequencing using a pair of specific primers, 27F (5′-AGAGTTTGATCMTGGCTCAG-3′) and 1492R (5′-GGYTACCTTGTTACGACTT-3′), as previously described (Chang et al., 2014). DNA sequencing were compared to reference sequences using the basic local alignment search tool of the National Center for Biotechnology Information database.

PCR and DNA sequencing of the topoisomerase gene

Isolates were screened for mutations in the gyrA, gyrB, parC, or parE genes by PCR using species-specific primers (Table 1). PCR products were sequenced for detection of nucleotide polymorphism. Primers were commercially synthesized by Genomics (New Taipei city, Taiwan). The reaction mixture (50 μl) contained 10 mM Tris–HCl (pH 7.5), 50 mM KCl, 1.5 mM MgCl2, 0.2 mM dNTPs, 10 pmol of the forward and reverse primer, 50 ng template DNA, and 0.8 U of Taq DNA polymerase (Applied Biosystems, Foster City, CA, USA). Amplification was carried out in a ProFlex PCR thermal cycler (Applied Biosystems, Foster City, CA, USA) with one initial denaturation step of 2 min at 95 °C; 40 cycles of a denaturing step of 15 s at 94 °C, an annealing step of 1 min at 48–50 °C with corresponding genes, and an extension step of 1 min at 72 °C; and a final elongation step of 5 min at 72 °C. All PCR products were processed for DNA sequencing (Genomics, New Taipei city, Taiwan) with the same PCR primer sets. Sequencing results in candidate genes from each isolate were compared with the respective reference sequences in the GenBank database (NCBI reference sequences: E. anophelis, NZ_CP007547.1; E. meningoseptica, NZ_CP016376.1; E. miricola, NZ_CP023746.1).

Table 1 Primer sequences used in this study.

Primer sequences used to amplify gyrA, gyrB, parC, and parE genes in Elizabethkingia spp.	
Primer name	Sequence (5′→3′)	Annealing temperature (°C)	Product size (bp)	
gyrA-E.species-F*	AGC CCG TTG TTT AAA TCC TGA A	50	743	
gyrA-E.species-R	CCC TGT TGG GAA GTC TGG TG	
gyrB-E.species-F	GAT AAT TTC CTT CAT AAA GAG CC	48		
gyrB-E.anophelis-R	CAT TGC CAT ACT GAG CTT GT	905	
gyrB-E.meningoseptica-R	TCG AAG TGT TTG CTT TGT CA	896	
gyrB-E.miricola-R	GCG TTG TCA TAC TGA ACT TG	903	
parC-E.species-F**	GCT CAG TAT GGC AAT GCT AAA A	50	785	
parC-E.species-R	TTG CTC TTA CCT TAC CGC CG	
parC-E.meningoseptica-F	TGA CCG GAT CAA CCG AAG TC	
parC-E.meningoseptica-R	CAG GTC GCC TGT TGT TTT GG	
parE-E.species-F	GTA TTC AGT TTA AAA GGT AAA CC	48		
parE-E.anophelis-R	GAA TAT ATT GGG CTT CGA CA	694	
parE-E.meningoseptica-R	ACT GAA CTT AGT TTG CCA TAA G	657	
parE-E.miricola-R	AGA AAT CGA CAT ATT CAG AGG T	683	
Primer sequences used for RAPD-PCR.	
Primer name	Sequence	Target	
OPA-10	GTG ATC GCA G	E. anophelis	
OPB-15	GGA GGG TGT T	E. meningoseptica	
E. miricola	
Primer sequences used for fluoroquinolone HRM analysis assays.	
Primer name	Sequence (5′→3′)	Annealing temperature (°C)	Product size (bp)	
gyrA-HRM-E.species-F	TGC CAG AAT TGT TGG AGA TG	50		
gyrA-HRM-E.anophelis-R	TAG CGC AGA GAC CAT GAC TG	102	
gyrA-HRM-E.meningoseptica-R	GTG CCA TAC GCA CCA TAG CA	83	
gyrA-HRM-E.miricola-R	CTG TGC CAT ACG CAC CAT AG	85	
Notes:

* gyrA-E.species-F and gyrA-E.species-R could amplify all Elizabethkinga species gyrA gene (including E. anohpelis, E. meningoseptica, and E. miricola).

** parC-E.species-F and parC-E.species-R could amplify both E. anohpelis and E. miricola parC gene.

High-resolution melting assay for gyrA mutation screening

Three different reverse primers and one common forward primer with homology to the Elizabethkingia spp. gyrA gene were designed (Table 1). PCR amplification was performed using the KAPA HRM FAST PCR Kit for preparing the following reaction: 20 μl reaction mix containing one μl template DNA (10 ng), eight μl PCR grade nuclease-free H2O, 10 μl KAPA HRM FAST Master Mix, two μl 25 mM MgCl2, and 0.5 μl of forward/reverse primer mix (10 μM each). The amplification and high-resolution melting (HRM) curve analyses were conducted on a LightCycler 96 instrument (Roche, Mannheim, Germany) using the following cycling conditions: initial activation at 95 °C for 2 min, 40 cycles at 95 °C for 10 s and at 60 °C for 30 s. The post-PCR melting curve was performed using temperatures between 65 and 95 °C in temperature increments of 0.3 °C.

RAPD-PCR and capillary gel electrophoresis analysis

RAPD-PCR was performed using primers (Table 1) described previously (Hsueh et al., 1996; Chiu et al., 2000). The reaction mixture (25 μl) contained 10 mM Tris–HCl (pH 7.5), 50 mM KCl, 2.5 mM MgCl2, 0.2 mM dNTPs, 15 pmol of the RAPD primer, 50 ng genomic DNA, and 0.8 U of DyNAzyme II DNA polymerase (ABI, ThermoFisher Scientific, Foster City, CA, USA). For every sample, each RAPD reaction was performed at least twice for each DNA extract. Amplification was carried out in a ProFlex PCR thermal cycler (Applied Biosystems, Foster City, CA, USA) with one initial denaturation step of 5 min at 95 °C; 40 cycles of a denaturing step of 1 min at 94 °C, an annealing step of 1 min at 36 °C, and an extension step of 2 min at 72 °C, and a final elongation step at 72 °C for 8 min.

After PCR amplification, the products were analyzed on Qsep100 DNA Analyzer (BiOptic, New Taipei City, Taiwan) according to the manufacturer’s instructions. PCR fragments were applied into a miniaturized single-channel capillary cartridge of the Qsep100 DNA-CE with separation buffer. The run was performed using a high-resolution cartridge with a sample injection protocol of eight kV for 10 s and separation at five kV for 300 s. The DNA alignment markers (20 bp, 1.442 ng/μl, and 5,000 bp, 1.852 ng/μl) and the DNA size marker (50–3,000 bp, 10.5 ng/μl) were obtained from BiOptic. Sample peaks were visualized using Q-Analyzer software (BiOptic, New Taipei City, Taiwan).

Molecular pattern analysis

Isolates were categorized as identical, similar or unrelated according to their PCR banding patterns. The data were analyzed using GelCompar II software (Applied Maths NV, Sint-Martens-Latem, Belgium). Dice similarity coefficients were calculated and clustering was done by unweighted pair group mean association.

Data analysis

Statistical significance was determined using Student’s t-test (GraphPad Prism. GraphPad Software Inc, San Diego, CA). Differences were considered statistically significant when P < 0.05.

Results

CIP and LVX susceptibility profiles of Elizabethkingia spp. isolates and corresponding resistance mutations

The 131 Elizabethkingia spp. isolates differed in their susceptibility to CIP and LVX (Fig. 1; Table 2); 91% and 77% were resistant to CIP and LVX, respectively. All E. meningoseptica isolates were resistant to CIP, whereas 44% were resistant to LVX; 73% of the E. miricola isolates were resistant to CIP and 27% were resistant to LVX. Most E. anophelis isolates were resistant to CIP and LVX (92% and 85%, respectively).

Figure 1 Fluoroquinolone MIC values of Elizabethkingia species.

(A) E. anophelis isolates (n = 111). (B) E. miricola isolates (n = 11). (C) E. meningoseptica isolates (n = 9). Each symbol (◊, □, ▵, ○) represents one isolate. CIP, ciprofloxacin; LVX, levofloxacin; S, susceptible; I/R, intermediate/resistant Susceptibility (≤ value), intermediate and resistance (≥ value) breakpoints defined by CLSI (2016): one, two and four μg/ml for CIP; two, four and eight μg/ml for LVX. **P < 0.01; ***P < 0.001; ****P < 0.0001.

Table 2 Antimicrobial susceptibility of ciprofloxacin/levofloxacin and mutation position detected in the gyrase or topoisomerase IV genes of Elizabethkingia species isolates.

Species	Number of isolate	MIC (μg/ml)	Mutation sites	
CIP*	LVX*	gyrA	gyrB	parC	parE	
Elizabethkingia anophelis	88	32–128	16–128	Ser83Ile	------------ No mutation ---------------	
6	32	8–64	Ser83Arg	------------ No mutation ---------------	
17	0.25–2	0.5	-------------- No mutation --------------------	
Elizabethkingia meningoseptica	5	32–64	32–64	Ser83Ile	------------ No mutation ---------------	
4	2	0.25–2	-------------- No mutation --------------------	
Elizabethkingia miricola	1	32	16	Ser83Ile	------------ No mutation ---------------	
2	32	4	Asp87Asn	------------ No mutation ---------------	
8	0.5–2	0.5	-------------- No mutation --------------------	
Note:

* CIP, ciprofloxacin; LVX, levofloxacin. Susceptibility (≤ value) and resistance (≥ value) breakpoints defined by CLSI (2016): one and four μg/ml for ciprofloxacin, two and eight μg/ml for levofloxacin.

A total of 101 (77%) Elizabethkingia spp. isolates had single-nucleotide mutations in the QRDR of the gyrA gene, whereas no mutations were found in the gyrB, parC, or parE gene of these isolates. In contrast, none of the 30 LVX-susceptible Elizabethkingia spp. isolates had mutations in the topoisomerase genes.

Among E. anophelis isolates with a gyrA gene mutation, 88 (93.6%) had a single-nucleotide mutation resulting in Ser83Ile amino acid substitution, whereas a different nucleotide mutation in six isolates resulted in Ser83Arg substitution. The most common single-nucleotide mutation encoding the Ser83Ile substitution was also found in E. meningoseptica and E. miricola isolates. Another single-nucleotide mutation in the gyrA gene, encoding an Asp87Asn substitution, was found in two E. miricola isolates. Our results indicate a strong correlation between the antibiotic susceptibility profiles of the clinical isolates and their mechanisms of fluoroquinolone resistance. The resistance against CIP and LVX in Elizabethkingia spp. is mainly mediated by a single-nucleotide mutation in the QRDR of the gyrA gene.

The 29 isolates without any mutation in gyrA, gyrB, parC, or parE were completely susceptible or had intermediate susceptibility to CIP (MIC, 0.25–2.00 μg/ml) and LVX (MIC, 0.25–2.00 μg/ml), whereas 102 isolates were fully resistant to CIP with a corresponding MIC range of 32–128 μg/ml and an LVX MIC range of 4–128 μg/ml (Table 3).

Table 3 Alterations in gyrA genes detected by HRM assay and confirmed by DNA sequence analysis in Elizabethkingia spp. Isolates.

Species	Number of isolate	Mutation detected by HRM	Confirmation by sequencing	
gyrA83	gyrA87	gyrA gene	
E. anophelis	94	Mutation	None	Ser83Ile/Ser83Arg	
17	None	None	No mutation	
E. meningoseptica	5	Mutation	None	Ser83Ile	
4	None	None	No mutation	
E. miricola	1	Mutation	None	Ser83Ile	
2	None	Mutation	Asp87Asn	
8	None	None	No mutation	

Rapid detection of gyrA mutations using the HRM assay

The results of the gyrA gene sequence analysis of Elizabethkingia spp. isolates for the identification of mutations in the QRDR were used to develop an HRM assay that can be used to rapidly scan clinical isolates for typical gyrA gene mutations in 131 isolates of Elizabethkingia species. The HRM assay successfully detected all gyrA mutations in this study, encoding the Ser83Ile, Ser83Arg, and Asp87Asn substitutions (Fig. 2). The HRM assay results for gyrA genotyping were in complete agreement with our DNA sequencing results without any exception (Table 3).

Figure 2 Representative HRM analysis of gyrA mutation and wild-type in Elizabethkingia species isolates.

(A) gyrA wild-type (n = 2) and gyrA mutation (n = 28) Elizabethkingia anophelis isolates. (B) gyrA wild-type (n = 5) and gyrA mutation (n = 4) Elizabethkingia meningoseptica isolates. (C) gyrA wild-type (n = 8) and gyrA mutation (n = 3) Elizabethkingia miricola isolates. WT, wild-type. Blue lines represent gyrA wild-type isolates, red lines and green lines represent gyrA mutation isolates, orange lines represent no template control.

RAPD-PCR typing of Elizabethkingia spp. isolates

The 131 Elizabethkingia spp. isolates were clustered into multiple pulsotypes defined by a similarity of ≥85% (Fig. 3). The widespread pulsotype clusters indicated a lack of temporal or spatial overlap among the infected patients during hospitalization. Specifically, pulsotypes of E. meningoseptica and E. miricola isolates harboring a gyrA mutation were found to be distributed among wild type clusters.

Figure 3 RAPD-PCR dendrogram of the Elizabethkingia spp. isolates investigated in this study.

(A) Clustering dendrogram of E. anophelis isolates (n = 111). (B) Clustering dendrogram of E. meningoseptica isolates (n = 9). (C) Clustering dendrogram of E. miricola isolates (n = 11). E.A: E. anophelis E.M: E. meningoseptica; E.m: E. miricola. Black triangles represent clusters with multiple isolates possessing the same gyrA mutations. Black circles represent monoisolate clusters with gyrA mutations. The dashed line represents the similarity level (85%) used in the clusters definition.

Discussion

Elizabethkingia spp. strains represent a group of emerging pathogens, causing infections that are associated with prolonged hospital stays and high mortality rates. In 2015–2016, there was an E. anophelis outbreak in Wisconsin, USA, that involved at least 63 patients and 18 deaths (Elbadawi et al., 2016). In addition, another outbreak in Illinois, USA, involving 10 cases with six deaths has also been reported in 2014–2016 (Navon et al., 2016). Globally, there are numerous sporadic E. meningoseptica nosocomial infection clusters and E. miricola infection case reports in medical centers including in Taiwan. Thus, pathogenic Elizabethkingia spp. strains appear to be opportunistic infectious agents associated with high mortality rates.

Quinolones underwent decades of development since the discovery of nalidixic acid in 1962, and quinolone resistance has also existed for decades. Recent studies described LVX-resistant E. meningoseptica bacteremia that is associated with an increase in mortality and prolonged hospital stays (Huang et al., 2017; Huang, Lin & Wang, 2018). Appropriate antibiotic use and an effective treatment regime are very important in fighting Elizabethkingia spp. infections. Using the broth microdilution method for MIC testing, we found differences in susceptibility to CIP and LVX among the Elizabethkingia spp. isolates. The discrepancy might be due to the different CLSI breakpoints, four μg/ml for CIP and eight μg/ml for LVX. Previously reported susceptibility profiles of E. anophelis isolates, including from the outbreak in Wisconsin in 2016, indicated that most isolates were susceptible to quinolones (Lau et al., 2016; Perrin et al., 2017). In sharp contrast, among our 111 E. anophelis isolates, only nine and 17 were found to be susceptible to CIP and LVX, respectively.

The genetic determinants of quinolone resistance have never been studied in Elizabethkingia spp. Our results revealed that certain single-nucleotide substitutions in gyrA conferred resistance to CIP and LVX in Elizabethkingia spp. The Ser83Ile substitution in GyrA protein was caused by the most prevalent mutation among all isolates, followed by the Ser83Arg or Asp87Asn amino acid substitutions caused by single-nucleotide mutations in E. anophelis or E. miricola. To our knowledge, this is the first report discussing genetic quinolone resistance determinants in Elizabethkingia spp.

Bacterial isolates carrying single alterations in QRDRs of DNA gyrase and topoisomerase IV typically exhibit reduced susceptibility to fluoroquinolones, which is considered as the first step in the development of full resistance (Hooper & Jacoby, 2017). The genetic basis for fluoroquinolone resistance appears to be additive, different combinations of distinct resistance mechanisms may result in different MIC (Conley et al., 2018). Other resistant mechanisms such as plasmid-mediated quinolone resistance might also be involved in the quinolone resistance mechanism (Yugendran & Harish, 2016). In our study, resistance to CIP and LVX was associated with single-nucleotide mutations in the QRDR of the gyrA gene in all Elizabethkingia spp. isolates causing low-level to high-level fluoroquinolone resistance. The level of fluoroquinolone resistance did not correlate with the type of mutation found in the gyrA gene. Other mechanisms typically implicated in fluoroquinolone resistance might be responsible for the differences in CIP and LVX MIC observed among the isolates. Changes in permeability and increased efflux pump activity along with plasmid-encoded resistance determinants cannot be excluded.

In this study, we also established a novel rapid HRM assay for detecting gyrA mutations in Elizabethkingia spp. The HRM results were in complete agreement with the DNA sequencing results, indicating that we developed a potentially useful adjunct test for the rapid detection of CIP and LVX resistance in Elizabethkingia spp.

Conclusions

Our findings demonstrated that the quinolone resistance in Elizabethkingia spp. is associated with mutations in the QRDR of the gyrA gene. However, the level of resistance to quinolones of Elizabethkingia spp. isolates could not be predicted based on the mutations identified in the gyrA gene. This study provided information for further research on the variations of the fluoroquinolone resistance mechanism and potential clinical guidance for infection management.

Supplemental Information

Supplemental Information 1 Raw data of our clinical isolates.

All elizabethkingia species collected in our study, with MIC values against ciprofloxacin and levofloxacin. We also calculated antimicrobial susceptibility percentage with susceptible/intermediate/resistant in all isolates.

Click here for additional data file.

Additional Information and Declarations

Competing Interests

Author Contributions

DNA Deposition

Data Availability

The authors declare that they have no competing interests.

Ming-Jr Jian conceived and designed the experiments, performed the experiments, analyzed the data, prepared figures and/or tables, authored or reviewed drafts of the paper, approved the final draft.

Yun-Hsiang Cheng performed the experiments, approved the final draft.

Cherng-Lih Perng contributed reagents/materials/analysis tools, approved the final draft.

Hung-Sheng Shang contributed reagents/materials/analysis tools, prepared figures and/or tables, authored or reviewed drafts of the paper, approved the final draft.

The following information was supplied regarding the deposition of DNA sequences:

Our sequences were aligned with the reference sequences via GenBank accession numbers described as follows:

E. anophelis, NZ_CP007547.1;

https://www.ncbi.nlm.nih.gov/nuccore/NZ_CP007547

E. meningoseptica, NZ_CP016376.1;

https://www.ncbi.nlm.nih.gov/nuccore/1153881081

E. miricola, NZ_CP023746.1;

https://www.ncbi.nlm.nih.gov/nuccore/NZ_CP023746.1

The following information was supplied regarding data availability:

The raw data are provided in the Supplemental Files.

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
