# Peer review of "Molecular typing and profiling of topoisomerase mutations causing resistance to ciprofloxacin and levofloxacin in Elizabethkingia species"

_PeerJ, doi:10.7717/peerj.5608_

## Round 0.1 · original submission · Major Revisions

As you can see, your paper was considered important and useful by both reviewers. Each of them, however, raised specific criticisms and made remarks that you should take into account when submitting a revised version. Please, indicate clearly in your rebuttal what has been changed (or not changed) as per the reviewers' comments.

Reviewer 1 ·

Basic reporting

no comment

Experimental design

no comment

Validity of the findings

no comment

Additional comments

Comments for the author:

This is an interesting study done by Shang and coworkers. Authors have done extensive work to understand the resistance in Elizabethkingia spp against ciprofloxacin and levofloxacin drugs. 131 Elizabethkingia spp isolates were used in this study. Authors have done extensive molecular biology, microbiology to identify the bacterial strains, mutations, and calculated the MIC values. They have also found that most of the mutations are detected in GyrA gene which is responsible for the drug resistance. The conclusions of the study are well stated and linked to the original research question. This study will be highly appreciated by mechanistic researcher who are trying to understand the drug resistance mechanism in Elizabethkingia spp against ciprofloxacin and levofloxacin. The study has been carried out with great care and caution, however, two small issues demand attention by the authors.

1. The last word of the title of the paper should be “species”

2. The font size of the Figure labels is small. Please increase the font size for the better visibility.


Clear and unambiguous; and professional English used throughout in the manuscript. I would recommend this important piece of work for the publication in PeerJ, after raised issues are taken care.

Reviewer 2 ·

Basic reporting

no comment

Experimental design

no comment

Validity of the findings

no comment

Additional comments

The article entitled "Molecular typing and profiling of topoisomerase mutations causing resistance to ciprofloxacin and levofloxacin in Elizabethkingia species", determined fluoroquinolone susceptibility profiles of clinical Elizabethkingi. These results provide information for further research on the variations of the resistance mechanism and potential clinical guidance for infection management. I recommend it for publication in PeerJ, but there are some parts to be improved and revised before acceptance:

>The quality of Fig1 is poor, please update this figure so that the readers can clearly understand it.

>Please label any significance analysis’s value on the figure(s).

>All figure’s legend must be described in detail.

>Fig2 (A, B, C) must be merged into one page.

>Fig3 (A, B, C) must be merged into one page.

>The authors should give the corresponding information about use of the sequences and how to construct the phylogenetic tree in Fig3.

---

## Round 0.2 · accepted · Accept

Both reviewers were happy with your revised version.

# Reviewer 1 ·

Basic reporting

no comment

Experimental design

no comment

Validity of the findings

no comment

Additional comments

I am happy with the revision authors have made. It is an excellent paper and worthy of publication in PeerJ.

Reviewer 2 ·

Basic reporting

no comment

Experimental design

no comment

Validity of the findings

no comment

Additional comments

The revised MS is OK for publication in PeerJ.